# Noise Enhancement of Neural Information Processing

**DOI:** 10.3390/e24121837

**Published:** 2022-12-16

**Authors:** Alain Destexhe

**Affiliations:** CNRS, Paris-Saclay Institute of Neuroscience (NeuroPSI), Paris-Saclay University, 91400 Saclay, France; alain.destexhe@cnrs.fr

**Keywords:** cerebral cortex, asynchronous states, information processing, computational models

## Abstract

Cortical neurons in vivo function in highly fluctuating and seemingly noisy conditions, and the understanding of how information is processed in such complex states is still incomplete. In this perspective article, we first overview that an intense “synaptic noise” was measured first in single neurons, and computational models were built based on such measurements. Recent progress in recording techniques has enabled the measurement of highly complex activity in large numbers of neurons in animals and human subjects, and models were also built to account for these complex dynamics. Here, we attempt to link these two cellular and population aspects, where the complexity of network dynamics in awake cortex seems to link to the synaptic noise seen in single cells. We show that noise in single cells, in networks, or structural noise, all participate to enhance responsiveness and boost the propagation of information. We propose that such noisy states are fundamental to providing favorable conditions for information processing at large-scale levels in the brain, and may be involved in sensory perception.

## 1. Introduction

Brain activity is subject to various sources of variability and noise, from the thermal noise present in ion channels, noisy membrane potential activity in single neurons, networks and circuits displaying highly irregular dynamics, up to the whole brain, where global electric and magnetic brain signals also display considerable amounts of noise. In this perspective article, we would like to explore internal sources of noise that are present at different scales, and what possible role they could have in neuronal computations.

At the single-cell level, neurons are subject to highly fluctuating and seemingly noisy conditions in vivo. This fact has been noted since the early days of the recording of brain electrical activity. The first intracellular recordings of cortical neurons in awake animals [1,2,3] showed that neurons are depolarized and subject to an intense and irregular synaptic bombardment of both excitatory and inhibitory inputs. This led to the proposal that cortical neurons function in a “high-conductance state” of neurons, which was inspired from early theoretical studies [4,5,6,7,8] (reviewed in [9]). The high-conductance state could be measured in a series of early experiments [6,10] which quantified the level of membrane potential ($V_{m}$), the amount of $V_{m}$ fluctuations and the total conductance of synaptic inputs, which are three of the fundamental parameters necessary to describe high-conductance states.

At the network level, the more recent spectacular progress in extracellular recording techniques for recording or imaging large numbers of neurons has established that the distributed activity of neurons in awake cortex is sustained, irregular and highly complex [11,12,13]. Such observations corroborate studies based on single-neuron measurements and models. In the present perspective, we would like to link these two levels, and propose that the synaptic “noise” present in neurons, the highly complex network activity and structure, all combine to confer interesting computational properties to these active networks.

## 2. Noisy Neurons

The first detailed measurements of synaptic noise in cortical neurons in vivo were performed in studies in cats [6,10], and are summarized in Figure 1. The three parameters mentioned above, the mean $V_{m}$ ($\mu_{V}$), the level of $V_{m}$ fluctuations ($\sigma_{V}$) and the effective membrane time constant ($\tau_{V}$) were measured. Note that the effective membrane time constant is given by the capacitance divided by the total conductance, so it is equivalent to measure the membrane conductance (or input resistance), as done in [6,10]. These measurements were performed in neurons recording in intact network activity in vivo, compared to the same neurons after total suppression of network activity using tetrodotoxin (TTX) microdialysis [10]. The measurements shown in Figure 1 were obtained by comparing periods of network activity (Up states) with the resting of the cell after TTX [6].

These measurements quantified, for the first time, that neurons in cortical neurons in vivo are subjected to intense synaptic activity, responsible for a depolarized $V_{m}$, intense $V_{m}$ fluctuations and a short time constant (high conductance), and that these conditions have a strong impact on single neurons. The measurements were incorporated in detailed computational models of neocortical pyramidal neurons [6] and showed that the sustained release of excitatory and inhibitory synapses in soma and dendrites could replicate all measurements, but only if a low level of correlation was introduced between synaptic release events. This low correlation matched the measurements made between pairs of neurons in awake animals [15].

It must be noted that not all measurements agree with these values. While some measurements in awake animals [16] or under anesthesia [6,10,17] do suggest a high-conductance state, some measurements in awake animals find that the conductance due to background activity seems negligible, or even negative, compared to states with quiescent activity (Down states) [18]. These differences were attributed to the presence of strong rectifying currents which mask the conductance of synaptic activity. This rectifying current was not present in the previous measurements, as shown by the linear V−I relations [6,16]. Note also that no computational model was proposed for the zero or negative conductance measurements, so this issue still requires further study.

Computational models were not only used to find plausible synaptic release conditions to explain conductance measurements, but they also can be used to infer computational properties of neurons in the presence of synaptic noise. A number of interesting properties have been found, and perhaps the most consequent one was that, with synaptic noise, cortical neurons present an enhanced responsiveness to synaptic inputs [14,19,20]. Due to noise, the response to inputs becomes non-determinsitic, or probabilistic, and the probability to evoke spikes by a given input can be boosted in the presence of noise. This phenomenon was also identified in other studies and was also called “gain modulation” [19]. A detailed series of dynamic-clamp experiments [20] showed the differential effect of the three parameters $\mu_{V}$, $\sigma_{V}$ and $\tau_{V}$. One of the main findings was that the membrane fluctuations, $\sigma_{V}$, were most effective in regulating the gain of the neural response, consistent with model predictions. It was also shown that synaptic noise modulates complex intrinsic properties such as bursting in thalamic neurons [21], resulting in a form of boosting of evoked responses. These findings motivated the development of theoretical models and further experiments to investigate this fluctuation-driven regime [17,22,23].

## 3. Noisy Networks

Multi-electrode or calcium imaging techniques have enabled the recording of large populations of neurons simultaneously, from animals to human. When performed during quiet wakefulness (under non-stimulated conditions), the distributed network activity appears sustained, irregular, and with very low levels of apparent synchrony. This network activity state is called “asynchronous irregular” (AI). This condition is illustrated in Figure 2A for awake human subjects recorded using multi-electrode Utah arrays [12].

These conditions can be replicated by simple network models. The AI state was actually first identified in computational models [26]. In that study, sparsely connected networks of integrate-and-fire excitatory and inhibitory neurons, were capable of displaying AI states, in addition to various other states, such as synchronized oscillations. AI states were also found in even simpler networks of binary neurons [27], or in more realistic networks of excitatory and inhibitory neurons matching the “regular-spiking” (RS) and “fast-spiking” (FS) intrinsic properties seen experimentally in cortical neurons (Figure 2B) [24]. In this case, more realistic conductance-based synaptic interactions were used, which allowed the model to be compared to the conductance measurements. It was found that networks of RS and FS neurons, with sparse connectivity, can generate AI states with membrane conductances consistent with in vivo measurements [24].

Again, computational models can be used to infer what advantageous properties AI states can offer. It is presently unclear if current-based or conductance-based interactions are important, but in conductance-based networks of RS-FS neurons, it was shown that AI states indeed can confer advantageous properties compared to other network states such as synchronized oscillations. In [25], it was shown that AI states provide enhanced responsiveness to external inputs, and that this property is responsible for an enhanced propagation of information in multi-layer networks (Figure 2C,D). This enhanced responsiveness at the network level is reminiscent of the enhanced response found at the cellular level (Figure 1C).

## 4. Structural Noise

Noise does not only apply to the activity of neurons or networks, but it can also be present in its structure. This structural noise, similar to the notion of “quenched disorder” in physics [28,29], is also present in the cerebral cortex which contains cells of very diverse shapes and sizes [30]. Figure 3 shows simulations of heterogeneous neural systems. In Figure 3A, networks of neurons were designed with different levels of cell-to-cell heterogeneity (which can be size, resting membrane potential, threshold, etc.) One can see that the homogeneous network is not the most responsive to external inputs, but here also, the presence of noise in the structure of the network is not detrimental but it seems to boost responsiveness [31,32]. Remarkably, the optimal responsiveness corresponds to the level of responsiveness measured experimentally in different preparations (Figure 3B). It was also found that there is a form of resonance to the level of heterogeneity (which can be seen in Figure 3B).

To investigate its impact at large scales, heterogeneous mean-field models were designed and could capture the responsiveness properties of heterogeneous networks [31]. Using these heterogeneous mean-fields, large-scale networks can be built, and in such large-scale networks, the system with moderate levels of heterogeneity was found to propagate information better than homogeneous systems (Figure 3C). This shows that microscopic heterogeneity, here from cell to cell, can have notable consequences at the large-scale level.

## 5. Discussion

In this perspective, we have shown three examples illustrating the strong impact that internal noise sources can have at different scales. At the cellular scale (Figure 1), we have shown the amount and impact of synaptic noise as seen from single neurons. In a sense, this noise can be seen as a feedback from network-level activity onto single cells, although of course, single cells collectively participate in setting up this network activity. At this single-neuron level, the experimental characterization [6,10] showed that synaptic noise is significant, and has a strong impact on neuronal parameters, such as the mean membrane potential ($\mu_{V}$), voltage fluctuations ($\sigma_{V}$) and the effective membrane time constant ($\tau_{V}$), which is linked to the total conductance in the cell. Playing on those parameters can change the position and the slope of the transfer function (Figure 1E). A particularly striking effect is that, for some input amplitudes, the response in the presence of noise is amplified compared to quiescent conditions (* in Figure 1E). This phenomenon of noise amplification bears some similarity to stochastic resonance phenomena [33], although the noise is here from network activity, so is “internal” to the system considered as a whole.

At the network level (Figure 2), we have illustrated that AI states are seen in awake human subjects (Figure 2A) and in awake animals [12], as well as in computational models where they are widely seen activity states (AI states constitute a large portion of the parameter space of these models) in sparsely connected spiking networks [26]. Remarkably, networks in AI states also can be more responsive to external inputs [25], similarly to the enhanced responsiveness seen at the cellular level. This enhanced responsiveness can also be seen by connecting multilayer networks (Figure 2C), where they can support the propagation of evoked activity across layers (Figure 2D). This also suggests that noise can be beneficial, where the “noise” is here the internal AI state exhibited by the system.

We also illustrated that noise does not need to be dynamical, but can also be structural (Figure 3). Here, networks made of neurons exhibiting cell-to-cell heterogeneity, for example in their intrinsic properties (size, excitability, resting level, etc.), can also be optimally responsive for non-zero levels of heterogeneity (Figure 3A). Interestingly, reporting the levels of heterogeneity measured experimentally falls in the region of maximal predicted responsiveness (Figure 3B). This cellular-level heterogeneity can have consequences at large scales, as shown by large networks of heterogeneous mean-field models (Figure 3C) [31]. Here again, the presence of noise at the structural level can lead to enhanced responsiveness and facilitation of information propagation.

Thus, based on those observations, the picture that emerges is that looking at the experimental conditions in single cell dynamics, in network dynamics, and in network structure, models predict that these different sources of noise tend to enhance the response of the system to external inputs and boost the propagation of information. This was proposed as the basis to explain why the awake and conscious brain is systematically associated with asynchronous and irregular activity states [25]. One may go a step further, and propose that the brain connectivity and cellular diversity are elements that are tuned to produce AI states of maximal responsiveness. Indeed, it was found that the human brain presents such an enhanced responsiveness in the waking state, compared to sleep or anesthesia [34,35,36]. Simulations of large-scale networks, based on the human connectome, showed that indeed, when the brain model is put in an asynchronous-irregular mode, it has an enhanced responsiveness compared to simulated slow waves [37,38].

Thus, cerebral cortex, by its structure with an extremely high cellular diversity and heterogeneity, combined with sparse and random connections [30], seems entirely consistent with what would be necessary to produce asynchronous-irregular states with optimal responsiveness. We have overviewed here how such states may provide enhanced responsiveness to external inputs, which may be consistent with their role in sensory perception. However, to investigate whether they are implicated in sensory awareness, or even consciousness, additional properties seem necessary. It was proposed that AI states can have the property that the information about an external input is immediately available to the whole network [25]. Such properties may provide interesting directions to explore why AI states are so systematically seen in the awake and active brain. Such complex activity states still have a lot to reveal, and understanding their properties and underlying mechanisms will require experiments with high spatial and temporal resolution, to characterize how AI states detect, propagate and communicate information across large scales and different brain areas. It will also require theoretical models to understand how information is represented in AI states, and how the interplay of the different scales organizes the underlying neuronal computations.

## Figures and Tables

**Figure 1 entropy-24-01837-f001:**
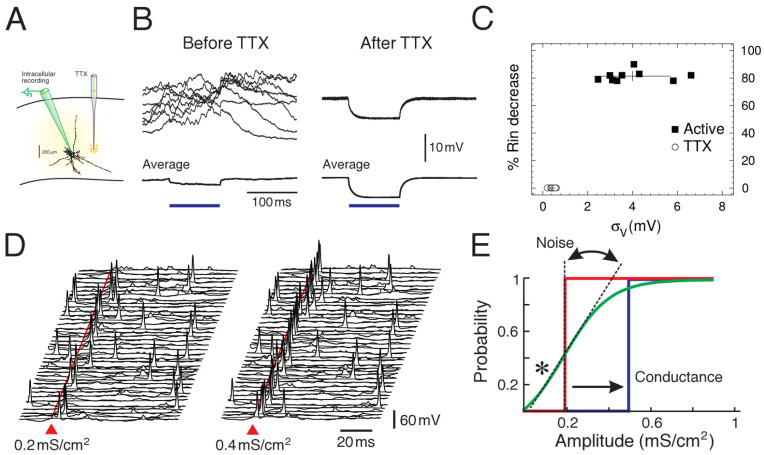
Synaptic noise in single neurons and enhanced responsiveness. (**A**) First measurements of synaptic noise in neuron. Neurons were recorded intracellularly in vivo, before and after microdialysis of tetrodotoxin (TTX). (**B**) Comparison of the membrane properties before and after TTX. Modified from [6,10]. (**C**) Measurements of mean voltage, voltage fluctuations ($\sigma_{V}$) and relative membrane resistance change ($R_{in}$). Modified from [6]. (**D**) Model of synaptic noise showing the probabilistic aspect of the spike response (two stimulus amplitudes shown, 40 trials each). (**E**) Effect of synaptic noise on the transfer function of the neuron (**D**,**E**) modified from [14].

**Figure 2 entropy-24-01837-f002:**
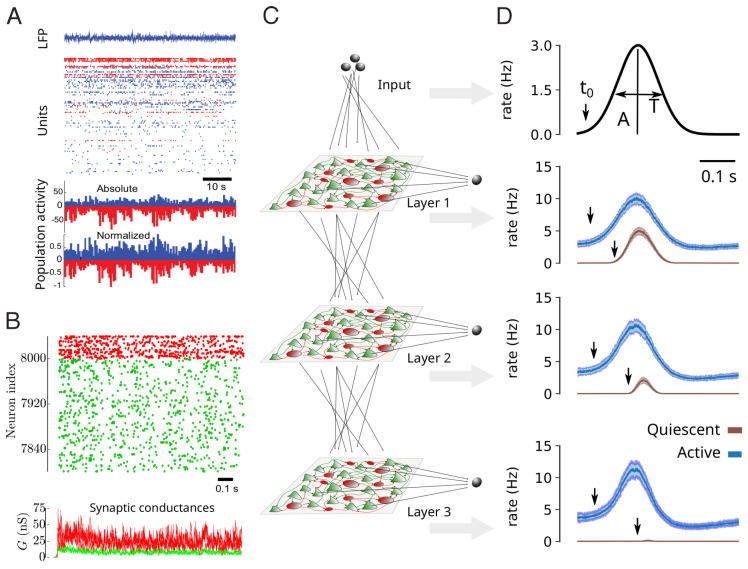
Asynchronous and irregular activity in cortical networks better propagate information. (**A**) Asynchronous-Irregular (AI) activity in an awake human subject recorded with a multi-electrode array. Excitatory (RS, blue) and inhibitory (FS, red) cells are shown, together with their average rate at the bottom. Modified from [12]. (**B**) AI states generated by AdEx networks, with conductance-based synapses. The bottom trace shows the total excitatory (green) and inhibitory (red) conductances in three example RS cells (superimposed traces). Modified from [24]. (**C**) Scheme of a multilayer arrangement of AdEx networks with excitatory inter-connections (dotted lines) and receiving input (top). (**D**) Input propagation across layers in AI states (blue curves), compared to non-propagation in quiescent states (brown). Modified from [25].

**Figure 3 entropy-24-01837-f003:**
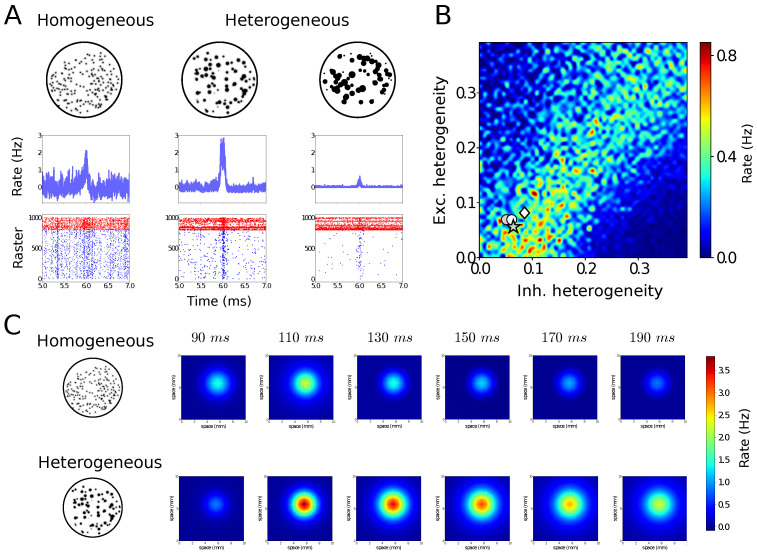
Heterogeneous networks can be highly responsive. (**A**) AdEx networks with different levels of heterogeneity, submitted to the same external input. A moderate level of heterogeneity presents the maximal response (middle). (**B**) Responsiveness in the plane of excitatory and inhibitory neuron heterogeneity. The maximum responsiveness (warm colors) occurs for intermediate levels of heterogeneity, and corresponds to the level of heterogeneity measured experimentally in cerebral cortex (white symbols). (**C**) Large-scale networks of heterogeneous mean-field units. When the units were based on heterogeneous networks (bottom), the propagating response was maximal. Modified from [31].

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
