# Peer review of "Noise Enhancement of Neural Information Processing"

_entropy, 2022, doi:10.3390/e24121837_

Round 1

Reviewer 1 Report

Overall this a good perspective and review paper on the sources of noise in neurobiological recordings and computational models.

My main comment is that, in my opinion, the introduction should be rewritten or restructured to more explicitly introduce the proposed thesis / argument that this paper is advocating.  The first paragraph is rather historical and somewhat narrow (the discussion of “high conductance” state of neurons) relative to the overall goal of unifying discussions of noise in structure, networks, and individual neurons.  This is especially the case, because this journal's readership is not simply neuroscience but rather a broader community interested in noise. 

If I had to summarize the paper, the central claim is that the brain – and specifically cortex - benefits from different implications of “noise” at different scales – the relationship of synaptic noise to neuron conductance state; the value of asynchronous irregular network activity in propagating information; and the value of structural heterogeneity vs homogeneity.  I feel like there is a stronger punchline buried in there; but that needs to be more explicitly called out in the introduction. 

I also think the paper would also benefit from a stronger “so what” argument at the end.  For instance, how does this perspective impact how we can look at brain regions with clear sources of heterogeneity (i.e., neurons born at different times in the hippocampus); or neural computing applications (artificial intelligence or neuromorphic computing) where historically the bias has been towards homogenous / noise-free implementations?   As it is, the forward direction is “we need to study it more”, which doesn’t really community why we need to study it more.

Minor comment

-          In the caption for Figure 2, the labels ‘FS’ and ‘RS’ are reversed.  FS = fast spiking (inhibitory) and RS = regular spiking (excitatory).  It should say “Excitatory (RS, blue) and inhibitory (FS, red) cells…”   The main text is correct (line 125).

Author Response

> Overall this a good perspective and review paper on the sources of
> noise in neurobiological recordings and computational models.

> My main comment is that, in my opinion, the introduction should be
> rewritten or restructured to more explicitly introduce the proposed
> thesis / argument that this paper is advocating.  The first
> paragraph is rather historical and somewhat narrow (the discussion
> of "high conductance" state of neurons) relative to the overall
> goal of unifying discussions of noise in structure, networks, and
> individual neurons.  This is especially the case, because this
> journal's readership is not simply neuroscience but rather a
> broader community interested in noise.  

The introduction was edited to better reflect this.

> If I had to summarize the paper, the central claim is that the
> brain - and specifically cortex - benefits from different
> implications of "noise" at different scales - the relationship of
> synaptic noise to neuron conductance state; the value of
> asynchronous irregular network activity in propagating information;
> and the value of structural heterogeneity vs homogeneity.  I feel
> like there is a stronger punchline buried in there; but that needs
> to be more explicitly called out in the introduction.  

The reviewer is right, and the Introduction was edited to better
reflect this.

> I also think the paper would also benefit from a stronger "so what"
> argument at the end.  For instance, how does this perspective
> impact how we can look at brain regions with clear sources of
> heterogeneity (i.e., neurons born at different times in the
> hippocampus); or neural computing applications (artificial
> intelligence or neuromorphic computing) where historically the bias
> has been towards homogenous / noise-free implementations?   As it
> is, the forward direction is "we need to study it more" doesn't 
> really community why we need to study it more.

Yes indeed, as suggested, the end of the discussion was expanded.

> Minor comment

> - In the caption for Figure 2, the labels "FS" and "RS" are
> reversed.  FS = fast spiking (inhibitory) and RS = regular spiking
> (excitatory).  It should say "Excitatory (RS, blue) and inhibitory
> (FS, red) cells" The main text is correct (line 125).

Many thanks for spotting this mistake !  It was corrected.

Reviewer 2 Report

This is a very interesting perspective on the functional significance of noise at the single-neuron and network levels. The structure and presentation of the paper convey reasonably well its main ideas. One very minor point (related to the use of the English language) is the following: it is "a large number of neurons", not "a large amount of neurons", unless the author means a large amount of neural mass. I do not have any further comments to make.      

Author Response

Thanks for the comment, yes indeed the first meaning was intended,
it was rewritten as suggested.